# Regulation of Vascular Calcification by Reactive Oxygen Species

**DOI:** 10.3390/antiox9100963

**Published:** 2020-10-08

**Authors:** Andrea Tóth, Enikő Balogh, Viktória Jeney

**Affiliations:** 1MTA-DE Lendület Vascular Pathophysiology Research Group, Research Centre for Molecular Medicine, Faculty of Medicine, University of Debrecen, 4032 Debrecen, Hungary; andrea.toth@med.unideb.hu (A.T.); balogh.eniko@med.unideb.hu (E.B.); 2Doctoral School of Molecular Cell and Immune Biology, Faculty of Medicine, University of Debrecen, 4032 Debrecen, Hungary

**Keywords:** vascular calcification, reactive oxygen species (ROS), vascular smooth muscle cells (VSMCs), osteochondrogenic transdifferentiation, Runx2

## Abstract

Vascular calcification is the deposition of hydroxyapatite crystals in the medial or intimal layers of arteries that is usually associated with other pathological conditions including but not limited to chronic kidney disease, atherosclerosis and diabetes. Calcification is an active, cell-regulated process involving the phenotype transition of vascular smooth muscle cells (VSMCs) from contractile to osteoblast/chondrocyte-like cells. Diverse triggers and signal transduction pathways have been identified behind vascular calcification. In this review, we focus on the role of reactive oxygen species (ROS) in the osteochondrogenic phenotype switch of VSMCs and subsequent calcification. Vascular calcification is associated with elevated ROS production. Excessive ROS contribute to the activation of certain osteochondrogenic signal transduction pathways, thereby accelerating osteochondrogenic transdifferentiation of VSMCs. Inhibition of ROS production and ROS scavengers and activation of endogenous protective mechanisms are promising therapeutic approaches in the prevention of osteochondrogenic transdifferentiation of VSMCs and subsequent vascular calcification. The present review discusses the formation and actions of excess ROS in different experimental models of calcification, and the potential of ROS-lowering strategies in the prevention of this deleterious condition.

## 1. Introduction

Ectopic calcification is the deposition of calcium (Ca) and phosphate (PO_4_^3−^, P)-containing hydroxyapatite crystals in any soft tissue. The most frequent type of soft tissue calcification is vascular calcification when Ca and P accumulate in the medial or intimal layers of the arteries [1]. For decades, vascular calcification was considered as a degenerative passive process related to aging and increased concentrations of P and Ca, a notion which was supported by the observation that elderly people and chronic kidney disease (CKD) patients are the most affected ones [2]. Vascular calcification is also frequently associated with various pathological conditions, such as diabetes, hypertension, atherosclerosis, osteoporosis, and rheumatoid arthritis [3,4]. 

The “passive” theory was challenged, and nowadays, it is widely accepted that vascular calcification is a cell-mediated highly regulated process that resembles, in many aspects, bone formation [5]. Vascular smooth muscle cells (VSMCs) present in the media layer of vessels play a central role in vascular calcification. VSMCs exhibit remarkable plasticity and respond to a variety of stimuli by changing their regular phenotype [6,7,8]. During vascular calcification, VSMCs undergo a phenotype switch, resulting in a cell type resembling osteoblasts or chondrocytes [9]. This phenotype switch is referred as osteochondrogenic transdifferentiation. The firstly described and the most-studied inducer of osteochondrogenic transdifferentiation of VSMCs is inorganic phosphate (Pi). The contribution of elevated Pi to CKD-associated vascular calcification has been confirmed undoubtedly [10,11]. Besides Pi, numerous triggers and inhibitors of such phenotype switching have been identified, and alterations in the balance of these pro- and anti-calcific stimuli are considered to eventually lead to ectopic mineral deposition [12]. 

Reactive oxygen species (ROS) are byproducts of aerobic metabolism. Physiological levels of ROS play a central role in redox signaling, whereas excessive ROS production causes oxidative stress, which is implicated in the initiation and progression of numerous diseases [13,14]. Accumulating evidence suggests that (i) vascular calcification is associated with elevated ROS production, and (ii) excess ROS play a pathophysiological role in the process of vascular calcification. In this review, we aim to summarize our current knowledge about the involvement of ROS in the development of vascular calcification.

## 2. Vascular Calcification

Vascular calcification is characterized by the accumulation of hydroxyapatite in the wall of large elastic arteries and coronary arteries. Vascular calcification has been related to increased risk of cardiovascular morbidities and complications [15], such as atherosclerotic plaque burden [16,17,18], myocardial infarction [19,20,21], coronary artery disease [22,23], ischemic stroke [21,24], postangioplasty dissection [25], and increased ischemic episodes in peripheral vascular disease [26]. Studies also indicate that coronary calcification may be predictive of or associated with sudden cardiac death [27,28]. Indeed, coronary calcification score has been shown to have a prognostic value for cardiovascular events comparable to that of the Framingham risk index [28].

Vascular calcification follows two distinct patterns, affecting both the intimal and medial layers of the arteries. Intimal calcification associates with atherosclerotic vascular disease, and is usually observed as spotty calcifications of the atherosclerotic plaques (reviewed in [29]). Recent investigations showed that while microcalcification is a key feature of unstable plaques, macrocalcification rather confers plaque stability, suggesting that the relationship of plaque calcification to plaque instability is very complex and not fully understood [30]. 

In contrast to intimal calcification, medial calcification, also known as Monckeberg sclerosis, is characterized by diffuse calcification of the media, particularly at the level of the internal elastic lamina. Media calcification is not necessarily accompanied by atherosclerosis but occurs predominantly in association with CKD and diabetes, and is linked to vessel stiffness, systolic hypertension, and increased pulse wave velocity, leading to increased diastolic dysfunction and eventually heart failure (reviewed in [29,31]).

CKD is associated with perturbations of phosphorous homeostasis. Accumulation of P starts relatively early in kidney disease, and positive phosphate balance occurs in the later stages (4 and 5) of CKD [32]. Observational studies revealed that hyperphosphatemia contributes to the high risk of death due to cardiovascular events observed in CKD [33,34]. Plenty of evidence supports that high serum P level predisposes CKD patients to metastatic calcification, the mechanism which is responsible for hyperphosphatemia-associated excess risk of cardiovascular disease in CKD [35,36]. 

Whilst originally thought to be a passive process, today, vascular calcification is considered as an active and tightly regulated mechanism that resembles, in many aspects, physiologic bone mineralization [1,9,31,37,38]. Several resident and circulating cell types can undergo osteogenic differentiation/transdifferentiation upon osteogenic stimulation and, therefore, can be involved in vascular calcification. These cells include vascular smooth muscle cells (VSMCs), smooth muscle- and endothelial cell progenitors, pericytes, and adventitial cells, as well as circulating mesenchymal and hematopoietic stem cells [39]. In this review, we will focus on osteogenic/osteochondrogenic transdifferentiation of VSMCs because this is the most extensively studied mechanism and the most widely accepted theory to explain vascular calcification. 

## 3. Osteochondrogenic Transdifferentiation of VSMCs

### 3.1. Inducers and Inhibitors of Osteochondrogenic Transdifferentiation of VSMCs

Osteochondrogenic transdifferentiation of VSMCs is the underlying cellular mechanism of vascular calcification that occurs when the balance between calcification inducers and inhibitors is impaired and the activity of inducers overwhelms the potential of inhibitors (Figure 1). One of the most potent inducers of osteochondrogenic transdifferentiation of VSMCs is elevated Pi, which is considered to be the most important pathophysiological trigger of media calcification in CKD [40,41,42]. Elevation of extracellular Ca levels has a synergistic effect on Pi-induced calcification [41], and a recent study found that dietary Ca correlates with vascular calcification and arterial stiffness [43]. Besides Pi and Ca, several inducers of VSMCs’ phenotype switch have been identified. Bone morphogenetic proteins (BMPs), members of the group of transforming growth factor beta (TGF-β) family, are multifunctional regulators of development and tissue homeostasis, and they were initially characterized as inducers of bone regeneration [44]. Among them, BMP2, BMP4, and BMP6 have been detected in calcified atherosclerotic lesions and they are thought to play a role in mineralization [45]. BMP2, in particular, has been shown to induce Pi uptake, phenotypic transition, and calcification of VSMCs [46]. Diabetes is associated with an increased prevalence of atherosclerotic vascular disease and cardiovascular mortality, and medial calcification appears to be a strong independent predictor of cardiovascular mortality in diabetic patients [47]. Correspondingly, elevated glucose and advanced glycation end products (AGEs) enhance the calcification of VSMCs [48,49]. Sterol-like molecules, such as dexamethasone, estradiol, and vitamin D3, are also implicated in the induction of vascular calcification [50,51,52]. Inflammation in general is associated with vascular calcification, in which the roles of several pro-inflammatory cytokines, including tumor necrosis factor alpha (TNF-α), oncostatin M, interleukin-1 beta (IL-1β), and IL-6, have been proposed [53,54,55,56,57,58]. Besides these inducers, local factors can also contribute to calcification (Figure 1). For example, oxidized low-density lipoprotein (oxLDL) that accumulates in the atherosclerotic lesion contributes to intimal calcification [59,60]. Additionally, recent studies showed that hypoxia, a condition where oxygen tension drops below its normal level in a particular tissue, triggers osteochondrogenic transdifferentiation of VSMCs and accelerates Pi-induced calcification [61,62].

Under homeostasis, the cardiovascular system is protected from elevated concentrations of serum Ca and Pi by a number of inhibitors that protect against abnormal hydroxyapatite deposition in soft tissues (Figure 1). These inhibitors include extracellular Ca-regulatory proteins such as matrix gla protein and fetuin-A [63,64]. Another group of inhibitors includes pyrophosphate (PPi) and extracellular nucleotides, as well as ATP and uridine triphosphate, which can serve as a source of pyrophosphate through ecto-nucleotide pyrophosphatase/phosphodiesterase-catalyzed cleavage [65,66]. Divalent cations, such as magnesium and iron, have also been shown to counteract with vascular calcification [67,68]. 

### 3.2. Transcriptional Regulation of VSMCs Osteochondrogenic Transdifferentiation

Osteochondrogenic transdifferentiation of VSMCs shares common signaling pathways with osteogenic differentiation of mesenchymal stem cells. Osteoblastogenesis is regulated by different secreted factors, including BMP-2, TGF-β, and members of the wingless/mouse mammary tumor virus integration site (Wnt) family. BMP-2 activates Smad signaling, whereas Wnts activate the canonical Wnt/β-catenin pathway through interaction with their receptor of the Frizzled family [69,70,71]. Activation of these signaling pathways leads to the upregulation and activation of certain osteoblast- or chondrocyte-specific transcription factors, such as Runx2, Sry-related HMG box-9 (Sox9), muscle segment homeobox homolog (Msx) 1 and 2, as well as osterix [69,71]. Among them, Runx2 seems to play a critical role in VSMC osteochondrogenic transdifferentiation, a notion which is supported by the finding that smooth muscle cell-specific Runx2 deficiency inhibits vascular calcification [72]. 

Upregulation of osteochondrogenic transcription factors leads to the expression of bone-specific proteins, including osteocalcin (OCN), osteopontin, and alkaline phosphatase (ALP) [50,73,74,75]. This phenotypic switch is completed by downregulation of VSMC lineage markers, such as smooth muscle α-actin (α-SMA) and SM-22α [50,73,74,75]. Similar phenotypic changes have also been observed in vivo in human calcified specimens, as well as in animal models of vascular calcification [76].

## 4. The Involvement of ROS in Vascular Calcification

### 4.1. Induction of Vascular Degeneration, Calcification and Osteochondrogenic Transdifferentiation by Excess Levels of ROS 

Oxidative stress and excessive production of ROS are important mediators of the osteochondrogenic transdifferentiation of VSMCs and have been associated with increased prevalence of vascular calcification. To prove the direct effect of ROS on vascular calcification, a novel laser-based method was established to generate focal ROS production in the aorta of rats [77]. The study revealed that local excessive ROS formation induces vascular degeneration and calcification [77]. Additionally, recent results showed that hypoxia, a local factor present in both intimal and medial calcification, induces osteochondrogenic transdifferentiation of VSMCs in a clearly ROS-dependent way [62].

### 4.2. ROS Production and Elimination in the Vasculature

There is a complex mechanism of production and elimination of ROS in the vasculature [78]. Vascular ROS are produced mainly by endothelial cells and VSMCs and are generated mostly by nicotinamide adenine dinucleotide phosphate (NADPH) oxidases (Nox). Nox enzymes are multi-subunit enzymes that catalyze the production of superoxide anion (O_2_^•−^) via one-electron reduction of oxygen, using NADPH as the electron donor. The Nox family comprises seven isoforms, Nox1-5, Duox1, and Duox2, from which Nox-1, Nox-2, Nox-4, and Nox-5 have functions in the vasculature [79]. In contrast to phagocytic Nox, which produce O_2_^•−^ in a burst-like manner upon activation, vascular oxidases continuously generate O_2_^•−^ in a slow and sustained fashion [79]. The produced O_2_^−•^ acts as an intracellular signaling molecule and regulates various vascular functions. 

Besides the vascular Nox enzymes, xanthine oxidase, the mitochondria, and uncoupling of endothelial NO synthase (eNOS) can produce O_2_^•−^ in the vasculature (Figure 2). Following its formation, O_2_^•−^ is converted to hydrogen peroxide (H_2_O_2_) by superoxide dismutase (SOD) enzymes. Transition metals, such as copper and iron ions (Cu^+^ and Fe^2+^), catalyze the conversion of H_2_O_2_ to hydroxyl radical (OH^•^) through the Haber–Weiss reaction. Once formed, OH^•^ can damage any cellular compartments due to its high reactivity. Antioxidant enzymes, such as catalase, glutathione peroxidase (Gpx), and peroxiredoxin (Prdx), convert H_2_O_2_ to water (Figure 2). 

### 4.3. Unfettered ROS Production in Vascular Calcification

Imbalanced ROS homeostasis and excessive ROS formation is associated with different vascular pathologies, including vascular calcification [80]. In line with this notion, elevated ROS production was detected in vivo in the calcifying aorta of CKD rats fed with a diet rich in Pi, Ca, and vitamin D [81], as well as around calcifying foci in the aortic valve of rabbits fed with a diet rich in cholesterol and vitamin D [82]. A growing body of evidence suggests that augmented ROS production enhances osteochondrogenic transdifferentiation of VSMCs. For example, hydrogen peroxide (H_2_O_2_) a cell-permeable ROS, and xanthine/xanthine oxidase that generates superoxide anion, have been shown to enhance osteochondrogenic transdifferentiation of VSMCs in vitro [83,84]. Enhanced calcification in the presence of H_2_O_2_ was associated with increased expression of osteogenic markers, including Runx2, OCN, and ALP, and decreased expression of the contractile VSMCs phenotype markers SM-22α and α-SMA [84]. Further study showed that homoarginine that induces oxidative stress in VSMCs also enhances high Pi-induced osteochondrogenic transdifferentiation of VSMCs and enhances vascular calcification in different mouse models [85]. Additionally, lipid oxidation products present in oxidized low-density lipoprotein also enhance ALP activity and calcification in calcifying vascular cells, a mechanism which might contribute to atherosclerosis-associated intimal calcification [59]. 

There are different sources of enhanced ROS production in the vasculature. Among them, Nox- and mitochondria-derived ROS have been associated with calcification. Agharazii et al. found increased expression of the Nox subunits p22(phox) and p47(phox) and decreased levels of the antioxidant enzymes SOD1, SOD2, Gpx1, and Prdx1 in the aorta of rats with CKD [81]. These alterations in ROS production and elimination lead to imbalanced ROS homeostasis and unfettered ROS formation. Several inducers of ROS production have been identified by in vitro studies under enhanced calcification conditions. For example, it has been shown that AGEs induce the expression of Nox-1, Nox-4, and p22(phox) and increase ROS production in VSMCs [86]. Silencing of Nox-4 and p22(phox) attenuated AGE-induced calcification, suggesting a causative role of Nox-4 in AGE-induced osteochondrogenic transdifferentiation of VSMCs [86]. The role of AGEs, receptor for AGEs (RAGE), and oxidative stress in inducing vascular calcification was further confirmed in vivo in a diabetic rat calcification model [87]. Direct activation of RAGE by the pro-inflammatory cytokine S100A12 augments CKD-triggered vascular calcification [88]. 

Liberman et al. showed that BMP-2 induces osteochondrogenic differentiation of VSMCs through increased Nox activity and enhanced ROS production [89]. Matrix vesicles derived from calcifying VSMCs also induce Nox-1 expression and calcification in VSMCs [90]. Matrix vesicles are extracellular vesicles with a high Ca^2+^ content which play a role in vascular calcification [91]. Recently, the role of Nox-5 has been uncovered in Ca^2+^ and matrix vesicle-induced vascular calcification [92]. In response to high extracellular Ca^2+^, VSMCs undergo a phenotype switch from a contractile to a synthetic phenotype, which is associated with increased ROS production and increased Nox-5 activity [92]. Modulation of Nox-5 expression itself regulates phenotypic marker expression of VSMCs; namely, Nox-5 overexpression decreases the levels of contractile markers, while decreased Nox-5 expression is associated with higher expression of contractile markers and reduced calcification potential [92].

Mitochondria can also serve as a source of excess ROS production in the vasculature, and elevation of mitochondria-derived ROS has been implicated in the pathomechanism of vascular calcification. Pi, the most relevant inducer of vascular calcification in CKD, has been shown to induce mitochondrial dysfunction, characterized by decreased mitochondrial membrane potential, mitochondrial fission, reduced ATP production, and increased generation of mitochondrial respiratory chain-derived ROS [93,94]. Accumulating evidence suggests a link between hypoxia and accelerated calcification. For example, arterial calcification is increased in patients suffering from asthma, chronic obstructive pulmonary disease, and obstructive sleep apnea [95,96,97]. Hypoxia partially inhibits the mitochondrial electron transport chain, leading to increased mitochondrial ROS production [98,99,100,101,102]. Recent evidence from Balogh et al. showed that hypoxia induces osteochondrogenic transdifferentiation of VSMCs, and highlighted the role of mitochondrial ROS in this hypoxia-induced response [62]. 

Overall, these results suggest the possibility that enhanced ROS production could be the common denominator that drives different signals towards the induction of osteochondrogenic differentiation of VSMCs [80]. 

## 5. Redox Regulation of Osteochondrogenic Signal Transduction Pathways

A growing body of evidence suggests the contribution of ROS in osteochondrogenic signal transduction pathways, and the involvement of redox signaling in osteochondrogenic transdifferentiation of VSMCs (Figure 3). 

### 5.1. BMP-2/Msx2/Wnt Signaling and Oxidative Stress

BMPs, named for their osteoinductive functions, are members of the transforming growth factor-β (TGF-β) family. BMP signaling plays a key role in embryonic skeletal development and postnatal bone homeostasis. BMPs transduce their signals via a tetrameric transmembrane receptor complex to both the canonical Smad-dependent, and the non-canonical, Smad-independent signaling pathways to regulate osteogenic commitment and differentiation of mesenchymal stem cells [103]. Besides its function in bone formation and bone homeostasis, BMP signaling has been implicated in various vascular pathologies, including vascular calcification [45,104]. Studies showed increased expression of BMP-2, BMP-4, and BMP-6 in areas of vascular calcification in atherosclerotic plaques [105,106]. Among them, the role of BMP-2 is the best studied in vascular calcification. BMP-2 induces the expression of osteogenic transcription factors Msx2 and Runx2, increases phosphate uptake, and enhances Pi-induced calcification in VSMCs [46,107]. 

There is a complex interplay between elevated ROS production and increased BMP-2 expression in triggering osteochondrogenic transdifferentiation of VSMCs. Dalfino et al. compared levels of BMP-2 and 8-Oxo-7,8-dihydro-2’-deoxyguanosine (8-OHdG), a marker of oxidative stress, in the serum of CKD patients (K-DOQI stage II or higher) and controls [108]. They found that both BMP-2 and 8-OHdG levels were higher in the CKD group than in the control, and BMP-2 levels directly correlated with 8-OHdG serum concentrations in the CKD patients [108]. Studies showed that H_2_O_2_ increases BMP-2 expression in endothelial cells, and that H_2_O_2_ enhances BMP-2-induced upregulation of ALP in VSMCs [108,109]. Moreover, Mandal et al. showed that BMP-2 induces a rapid generation of ROS through the activation of Nox-4 in pre-osteoblasts, and that BMP-2-induced ROS generation is essential for osteoblast differentiation [110]. 

Msx2 is a key factor involved in transcriptional programming of osteoblastic lineage development, and appears to also be a critical transcription factor driving BMP-2-mediated vascular calcification [111]. Shao et al. showed that Msx2 promotes vascular calcification through the activation of Wnt/β-catenin signaling [112]. 

Wnt signaling is indispensable for embryonic development through regulating axis patterning, cell fate decisions, cell proliferation, and organized cell migration [113]. Wnt signaling is highly complex due to the large number of ligands and receptors involved in the Wnt signal transduction and the variety of intracellular responses provoked by the receptor–ligand interactions [113]. Activation of the canonical Wnt pathway is initiated by the binding of classical Wnt proteins to the Frizzled/LDL receptor-related protein (Fzd/LRP) receptor and results in cytosolic accumulation and nuclear translocation of β-catenin. Inside the nucleus, β-catenin interacts with T cell transcription factor (TCF) and lymphoid enhancer factor (LEF) and induces transcription of β-catenin target genes [113]. The noncanonical (Wnt/Ca^2+^) pathway is activated by the binding of a Wnt-protein ligand to a Fzd family receptor, leading to activation of heterotrimeric G protein and regulation of intracellular Ca level [114].

During skeletal bone formation, paracrine epithelial–mesenchymal and endothelial–mesenchymal interactions control the osteochondrogenic differentiation of multipotent mesenchymal stem cells. The paracrine signals are mediated by proteins belonging to the BMP and Wnt superfamilies. Growing evidence suggests that this paracrine mechanism is involved in the osteochondrogenic reprogramming of arterial cells during vascular and valve calcification [112]. Shao et al. showed that overexpression of Msx2 in mesenchymal cells upregulated Wnt3a and Wnt7a but downregulated expression of Dickkopf-related protein 1 (Dkk1) that serves as an inhibitor of the canonical Wnt signal transduction pathway [112]. 

Canonical Wnt signaling leads to nuclear translocation of β-catenin and transcriptional activation of certain genes. Promoter reporter and chromatin immunoprecipitation assays showed that β-catenin-induced activation of Pit1, a type III sodium-dependent phosphate cotransporter, is involved in high Pi-induced calcification of VSMCs [115]. Furthermore, the contribution of the canonical Wnt signal transduction pathway in Pi-induced calcification was proven in vivo with the use of a rat model of CKD with reduced β-catenin expression [115]. Cai et al. showed that WNT/β-catenin signaling triggers osteogenic transdifferentiation and calcification of VSMCs via directly modulating Runx2 gene expression [116]. In this in vitro study, the authors showed that a high Pi level triggers the formation of two forms of active β-catenin, dephosphorylated on Ser37/Thr41 and phosphorylated on Ser675 sites, and activation of β-catenin was associated with increased Runx2 expression [116]. On the other hand, blockade of WNT/β-catenin signaling with an inhibitor or Dkk1 protein inhibited Runx2 induction by high Pi levels [116]. Deng et al. showed that inhibition of the WNT/β-catenin signal transduction pathway by secreted Frizzled-related protein 5 decreases high Pi-induced calcification in VSMCs [117]. Growing evidence suggests that the Wnt/β-catenin pathway is regulated at least in part by Nox-derived ROS, but further studies are needed to establish the role of ROS-induced Wnt signaling in vascular calcification [118]. 

### 5.2. Hypoxia/HIF-1 Signaling and Oxidative Stress

Oxygen is a fuel for cellular respiration and many other vital functions in most organisms, therefore adaptive mechanisms are required to maintain oxygen homeostasis. All nucleated cells are able to sense and respond to hypoxia through the activation of hypoxia inducible factors (HIFs) [119,120,121]. The HIF-1 pathway is the master regulator of cellular and systemic homeostatic responses to hypoxia. 

HIF-1 is a heterodimeric basic helix-loop-helix-PAS domain transcription factor, composed of an oxygen-sensitive alpha subunit (HIF-1α) and a stable beta subunit [122]. Under normoxia, HIF-1α is constantly synthesized, hydroxylated on two conserved proline residues by prolyl hydroxylase enzymes (PHDs), ubiquitinated by the von Hippel–Lindau E3 ubiquitin ligase, and targeted for 26 S proteasomal degradation. Under hypoxia, oxygen-dependent PHDs are inactive, leading to HIF-1α stabilization and dimerization with the beta subunit [123]. The heterodimer translocates into the nucleus, binds to cis-acting hypoxia response elements (HREs) in HIF-1 target genes, recruits coactivator molecules, i.e., p300 and cyclic adenosine monophosphate (cAMP) response element-binding protein and the complex activates the transcription of over 100 downstream genes [123]. 

HIF-1-induced signaling triggers adaptation mechanisms, including, but not limited to, angiogenesis, vascular reactivity and remodeling, and metabolic alterations, such as upregulation of glucose uptake and glycolysis, to foster survival in a hypoxic condition [124]. As part of the hypoxic response, cells reduce the oxygen consumption of mitochondria through upregulation of pyruvate dehydrogenase kinase 1 that leads to inactivation of pyruvate dehydrogenase, which prevents the conversion of pyruvate to acetyl-CoA and thus attenuates the entry of pyruvate to the mitochondrial Krebs cycle [125]. 

Intracellular ROS production also changes during hypoxia, but the direction of this change and the origin of the ROS (mitochondrial or NADPH oxidase) remain controversial [126]. Based on the assumption that ROS production requires oxygen, several studies supported the idea that ROS production is decreased under hypoxic conditions [126]. Other studies showed the opposite and provided evidence that hypoxia elevates mitochondrial ROS production through partial inhibition of the mitochondrial electron transport chain [98,99,100,101,102]. Recent evidence suggests that upon hypoxia, mitochondria-derived elevated ROS production stabilizes HIF-1α, and therefore, elevated ROS production plays a critical role in hypoxia-driven cellular responses [101,102,127]. 

Growing evidence suggests a link between hypoxia and vascular calcification. In line with this notion, accelerated vascular calcification was observed in patients with asthma, chronic obstructive pulmonary disease, and obstructive sleep apnea, conditions which are accompanied by hypoxia [95,96,97]. Hypoxia is also present in both characteristic types of vascular calcification affecting medial and intimal layers of the vasculature.

Medial arterial calcification is common in CKD and is associated with poor clinical outcomes. Recent studies revealed that renal tissue hypoxia is present in CKD, and hypoxia is the driving force of the pathogenesis of CKD, triggering renal fibrosis and contributing to the development of anemia, inflammation, and aberrant angiogenesis [128,129,130]. CKD-associated hypoxia is not strictly localized to the renal tissue, but is also present in the vasculature [61].

Intimal calcification is associated with atherosclerosis, a pathology characterized by sub-endothelial lipid accumulation, inflammation, and fibrosis. Hypoxia of the mid-region of the atherosclerotic plaques was demonstrated in various animal models, as well as in human carotid lesions [131,132,133,134,135]. Because of hypoxia, neovascularization—growth of capillary-like microvessels into the thickened media and intima—occurs, which was considered as a prominent feature of advanced atherosclerotic plaques [136]. Recent evidence suggests that in fact, hypoxia and subsequent neovascularization are present in early atherosclerotic lesions [137], particularly when the thickness of the tunica intima exceeds the maximum diffusion distance of oxygen, which is about 200–250 µm [138,139,140]. Animal studies revealed that hypoxia accelerates the progression of atherosclerosis [141,142]. Plaque hypoxia has been implicated in plaque neovascularization, altered metabolism, increased lipid accumulation, enhanced inflammation, and augmented proteolysis [143].

Some studies showed an association between the levels of HIF-1α and vascular calcification. For example, a correlation between plasma HIF-1α levels and coronary artery calcification has been shown in patients with type 2 diabetes [144]. Another study revealed that HIF-1α co-localizes with areas of calcification in stenotic valves [145]. Additionally, calcification-promoting factors were detected in patients with pulmonary arterial hypertension, a disease characterized by intense proliferation of pulmonary artery smooth muscle cells, pulmonary vascular remodeling, elevated pulmonary arterial pressure, vascular resistance, and hypoxia [146]. 

The fact that hypoxia and calcification coincide does not necessarily mean that hypoxia has a causative role in vascular calcification. This question was addressed in a few recent publications. Ruffenach et al. showed that pulmonary arterial hypertension is associated with increased pulmonary arterial calcification and elevated expression of Runx2 in the lungs of patients with pulmonary arterial hypertension [146]. Moreover, using in vitro gain- and loss-of-function approaches, they demonstrated that sustained Runx2 expression activates HIF-1α, which eventually leads to the transdifferentiation of pulmonary smooth muscle cells into osteoblast-like cells [146]. Mokas et al. investigated the effect of hypoxia on high Pi-induced osteogenic differentiation and calcification of VSMCs [61]. They showed that hypoxia largely intensified Pi-induced osteogenic transdifferentiation and calcification of VSMCs, a mechanism which could be relevant in mineral imbalance-induced calcification in patients with CKD [61]. Recently, Balogh et al. provided evidence that hypoxia is a bona fide trigger of VSMC calcification [62]. They showed that exposure of VSMCs to sustained hypoxia induces osteochondrogenic reprogramming characterized by increased expression of the osteochondrogenic master transcription factors Runx2 and Sox9 [62]. Interestingly, long-term exposure of VSMCs to hypoxia triggered mineralization of the extracellular matrix of VSMCs as well [62]. They provided evidence that unfettered production of ROS by the mitochondria plays a critical role in hypoxia-induced VSMC phenotype switching and calcification [62]. 

### 5.3. PERK/eIF2α/ATF4/CHOP Pathway

The endoplasmic reticulum (ER) is the first organelle of the secretory pathway where folding, posttranslational modification, and assembly of secreted and transmembrane proteins occur. Perturbations of ER functions cause ER stress, characterized by the accumulation of misfolded proteins in the ER [147]. To cope with the deleterious effects of ER stress and proteotoxicity, cells have evolved a protective strategy, referred to as the unfolded protein response (UPR). The UPR is a concerted and complex pro-survival response with the role of reducing the accumulation of misfolded proteins and restoring the normal ER function [148]. However, chronically persisting ER stress or failure of the adaptive response activate a terminal UPR program leads to apoptotic cell death [148]. Genetic mutations, environmental factors, and aging can trigger chronic ER stress and malfunction of UPR signaling, which are emerging as important contributors of the pathomechanism of various diseases, such as diabetes, inflammation, cancer, and neurodegenerative and cardiovascular disorders [149].

Increased load of unfolded proteins in the ER is sensed by three ER transmembrane receptors: pancreatic ER kinase (PKR)-like ER kinase (PERK), activating transcription factor 6 (ATF6), and inositol-requiring enzyme 1 (IRE1), defining the three branches of the UPR [148]. Under ER homeostasis, the ER stress sensors are kept in an inactive state through their association with the ER chaperone Grp78 [148]. Accumulation of unfolded proteins in the ER triggers dissociation of Grp78 from the ER stress receptors, allowing their activation [148]. Activation of PERK leads to phosphorylation of eukaryotic initiation factor 2α (eIF2α) and subsequent blockade of protein expression [148]. Furthermore, eIF2α phosphorylation induces the expression of activating transcription factor 4 (ATF4), which translocates to the nucleus and induces transcription of genes involved in restoration of normal ER function [148]. Activation of ATF6 induces the expression of ER chaperones and X box-binding protein 1 (XBP1), another transcription factor [148]. For the production of active XBP1 protein, XBP1 mRNA must be sliced, which is carried out by IRE1 [148]. Active XBP1 translocates to the nucleus and controls the transcription of further chaperones and genes involved in protein degradation [148]. 

ER stress and the UPR are known to play a role in cardiovascular pathologies and growing evidence suggests the critical involvement of the PERK/eIF2α/ATF4 branch of the UPR signaling pathway in vascular calcification [150]. In line with this notion, Duan et al. showed that nicotine- and vitamin D-induced vascular calcification was associated with increased expression of the ER stress markers Grp78 and Grp94 in the aorta of rats. Additionally, markers of ER stress-induced apoptosis, such as CCAAT-enhancer-binding protein homologous protein (CHOP) and caspase 12, were also elevated, suggesting that the terminal UPR program has been activated in the calcifying aorta [151]. A subsequent study proved a critical role of ATF4 as the mediator of ER stress-induced osteochondrogenic differentiation of VSMCs in vitro, and nicotine- and vitamin D-induced aorta calcification in vivo [152]. Furthermore, Masuda et al. showed the activation and the critical contribution of the PERK/eIF2α/ATF4/CHOP pathway in diverse in vitro and in vivo models of vascular calcification [153,154]. 

Protein folding is a redox-dependent process, and growing evidence suggests that oxidative stress and ER stress are intimately interrelated [155,156]. Studies showed that the association between oxidative stress and ER stress is not only coincidental, but that excessive ROS production during ER stress is an integral component of the UPR and can be involved in the propagation of both pro-survival and pro-apoptotic ER stress responses [157,158]. The major mechanism of ROS production in the ER is the formation of protein disulfide bonds, a process which is catalyzed by ER oxidoreductin 1 (Ero1) and its thiol redox partner protein disulfide isomerase. These thiol–disulfide exchange reactions are coupled to electron transfer from reduced Ero1 to oxygen, thus generating hydrogen peroxide [159]. Besides this, ER is involved in the assembly of Noxs that—besides the mitochondria—serve as major sources of ROS [160].

### 5.4. Nuclear Factor Kappa B (NF-κB) Pathway

The transcriptional response in innate and adaptive immunity is dependent on the activation of various transcription factors, including NF-κB. Since its discovery [161], the NF-κB signal transduction pathway has been a subject of extensive research, and many excellent reviews have been published on it [162,163,164]. NF-κB is the general name for a family of transcription factors which has five members: NF-κB1 (also named p50), NF-κB2 (also named p52), RelA (also named p65), RelB, and c-Rel, which mediate transcription of target genes by binding to a specific DNA element [162,163,164]. In resting situation, the NF-κB dimer is inactive by an inhibitory protein, the inhibitor of κB (IκB). Activation of NF-κB involves two major signaling pathways—the canonical and the noncanonical pathways [165]. 

The canonical NF-κB pathway responds to diverse stimuli, including ligands of various cytokine receptors, pattern recognition receptors, and TNF receptor superfamily members, as well as T-cell receptor and B-cell receptor. The primary mechanism for canonical NF-κB activation is the proteosomal degradation of IκBα via its site-specific phosphorylation by a multi-subunit IκB kinase (IKK) complex that is composed of two catalytic subunits, IKKα and IKKβ, and a regulatory subunit named NF-κB essential modulator, or IKKγ [162,163,164]. Once IκBα is degraded, the p65/p50 heterodimer translocates into the nucleus. IKK1 has been also shown to be involved in a noncanonical NF-κB pathway [162,163,164]. This pathway regulates the ubiquitin-mediated processing of p100, which modulates the levels of p52/relB heterodimers. 

In contrast to the classical p65/p50 dimers, p52/relB dimers do not associate with IκB proteins, but they are located in the cytoplasm as a p100/relB dimer. The processing of p100 to p52 in this alternative NF-κB pathway is dependent on IKK1, and the resulting p52/relB complex can translocate into the nucleus where it regulates the transcription of NF-κB-dependent genes [162,163,164]. Several signals have been shown to activate the noncanonical pathway, such as lymphotoxin-β, B cell–activating factor, CD40 ligand, lipopolysaccharide (LPS), and receptor activator of NF-κB.

Activation of the NF-κB signal transduction pathway has been associated with vascular pathologies. Intensive research has focused on the role of NF-κB in the regulation of endothelial cell activation and inflammatory cell responses, particularly in the pathogenesis of atherosclerosis [166]. Besides that, recent studies showed the importance of the NF-κB signal transduction pathway in regulating VSMC functions, such as response to injury, pro-inflammatory activation, and remodeling [167]. Besides these effects, it has been proven that activation of the NF-κB pathway plays a critical role in high Pi-induced osteochondrogenic transdifferentiation of VSMCs and vascular calcification [168]. It has been previously shown that unfettered production of ROS can activate the NF-κB pathway. Having established that high Pi induces excessive ROS formation and osteochondrogenic transdifferentiation of VSMCs, Zhao et al. investigated the role of the NF-κB pathway in this context [93]. They showed that high Pi increases mitochondrial ROS production in VSMCs and induces IKKβ phosphorylation and IκBα degradation, leading, eventually, to the translocation of p65 to the nucleus [93]. Knockdown of the endogenous p65 level or overexpression of IκBα greatly reduces high Pi-induced extracellular matrix calcification of VSMCs, indicating that NF-κB signaling is involved in this process [93]. Later, Zhao et al. showed that activation of the NF-κB pathway inhibits the expression of ankylosis protein homolog, a transmembrane protein that controls pyrophosphate efflux of VSMCs. Because pyrophosphate is a potent inhibitor of calcification, this mechanism can contribute to enhanced calcification upon NF-κB activation [93]. Importantly, smooth muscle cell-targeted inhibition of NF-κB decreases Pi-induced calcification in mice with CKD [169]. Activation of NF-κB promotes VSMC calcification, in part, by inducing Msx2 expression and up-regulation of Runx2 and its target, ALP [170]. 

Besides Pi, several other inducers of vascular calcification act through the NF-κB signal transduction pathway. For example, AGEs promote calcification of VSMCs via activation of NF-κB, which was evidenced by an experiment in which knockdown of p65 suppressed the AGE-induced increase in calcification [171]. Another example is oxLDL that induces calcification of VSMCs in a toll-like receptor 4- and NF-κB-dependent manner [172]. Additionally, a recent work of Voelkl et al. revealed that NF-κB plays a critical role in serum- and glucocorticoid-inducible kinase 1-induced calcification of VSMCs [173]. 

### 5.5. Mitogen-Activated Protein Kinase (MAPK) and the PI3K/Akt Pathways

Both MAPK and PI3K/Akt pathways are activated by various external stimuli, including growth factors, hormones, and stress. MAPKs are classified into several subfamilies, including extracellular signal-regulated kinases 1 and 2, p38 MAPK, c-Jun NH2-terminal kinase/stress activated protein kinase, ERK3/4, and ERK5. Activation of MAPKs leads to phosphorylation of downstream cytosolic regulatory proteins and many nuclear transcription factors, such as c-Jun, ATF2, ETS Like-1 protein, CHOP, cAMP response element-binding protein, and myocyte enhancer factor-2. The phosphorylation of intracellular proteins and transcription factors by MAPKs and PI3K/Akt leads to activation of several genes involved in growth, survival, and differentiation. Additionally, the MAPK and PI3K/Akt pathways are regulated by ROS and interconnected with other signal transduction pathways, such as NF-κB or HIF-1 signaling [174]. 

Radcliff et al. first showed the involvement of MAPK and PI3K/Akt pathways in insulin-like growth factor-induced osteochondrogenic transdifferentiation of VSMCs [175]. Moreover, evidence shows that elevated Pi induces both MAPK and PI3K/Akt pathways in diverse human cells [176]. Further triggers of vascular calcification, such as oxLDL, high glucose, and H_2_O_2_, have been shown to induce osteochondrogenic transdifferentiation of VSMCs via the activation of both MAPK and PI3K/Akt signaling pathways [84,177,178,179].

## 6. Controlling ROS Production as a Therapeutic Approach to Prevent Vascular Calcification 

In recent years, the involvement of excess ROS production in the regulation of phenotype switching of VSMCs from contractile to osteogenic became more and more evident. This raised the possibility of the use of ROS scavengers/antioxidants as therapeutic interventions to prevent and/or cure vascular calcification. 

Exposure of VSMCs to calcification inducers, such as high Pi, oxLDL, or hypoxia, increases ROS production. Inhibition of mitochondrial ROS production has been shown to inhibit high Pi-, oxLDL-, or hypoxia-induced osteochondrogenic transdifferentiation of VSMCs and vascular calcification in experimental systems [62,93,180]. Additionally, selective inhibition of Noxs also attenuates vascular calcification [90,181,182]. These results suggest that mitochondria- as well as Nox-derived ROS play a significant role in the phenotype switch of VSMCs and in the process of vascular calcification. 

Numerous natural and synthetic ROS scavengers have been tested in experimental settings as to whether they inhibit osteochondrogenic transdifferentiation of VSMCs and attenuate vascular calcification. A detailed review on this topic has been published recently [183]. Some naturally occurring plant compounds, such as quercetin, puerarin, apocynin, and rosmarinic acid, have been reported to possess anti-calcification properties through ROS scavenging [94,184,185,186,187]. Multiple medicines, such as simvastatin, a drug used to lower LDL cholesterol, and metformin, the first-line medication for the treatment of type 2 diabetes, also exhibit antioxidant properties and are effective against vascular calcification in experimental settings [188,189]. Synthetic antioxidants, such as N-acetyl cysteine or Tempol, have been shown to inhibit osteochondrogenic transdifferentiation of VSMCs and attenuate vascular calcification [62,190]. 

Mammalian cells have evolved endogenous protective mechanisms that counteract ROS generation. Nuclear factor erythroid 2-related factor 2 (Nrf2), is a key regulator of such protective responses, and recent evidence suggests that activation of Nrf2 may be beneficial in attenuating vascular calcification [191]. In the canonical Nrf2 pathway under normal conditions, Nrf2 is suppressed by Kelch-like ECH-associated protein 1 (Keap1), which leads to ubiquitylation and proteasomal degradation of Nrf2 [192]. Changes in the intracellular redox balance trigger inactivation of Keap1, leading to dissociation and nuclear translocation of Nrf2. In the nucleus, Nrf2 binds to antioxidant response elements (ARE) and increases transcription of various antioxidant and anti-inflammatory genes [193]. Different agents, such as dimethyl fumarate, resveratrol, rosmarinic acid, or hydrogen sulfide, have been shown to inhibit vascular calcification through stimulation of Nrf2 activity [185,194,195,196]. Yao et al. showed that overexpression of Nrf2 protects VSMCs from high Pi-induced calcification [197]. This effect is attributed to the activation of the Nrf2-ARE signaling pathway, leading to the induction of autophagy in VSMCs [197]. Additionally, Wei et al. found that silencing of Nrf2 increases VSMC calcification through elevated ROS production [198].

## 7. Conclusions

Coronary artery calcium score, a measurable indicator of vascular calcification, is an independent predictor of major adverse cardiac events in both the general population and in patients with coronary artery disease. Additionally, calcification of heart valves is the major cause of valve insufficiency.

Far from being a passive process, vascular calcification is an active and regulated process with remarkable complexity that involves numerous mechanisms that resemble bone formation. The key event in vascular calcification is the phenotype switch of vascular smooth muscle cells from a contractile to an osteochondrogenic phenotype. Numerous inducers of this osteochondrogenic phenotype switch trigger excess ROS production, and ROS have a causative role in the transdifferentiation process through their interactions with some major osteochondrogenic signal transduction pathways. 

So far, no therapeutic intervention exists that would specifically target vascular calcification. To develop an effective and specific anti-calcification therapy, we need further understanding of the molecular mechanism of the phenotype shift of VSMCs. Re-balancing the oxidative state of the vasculature may be beneficial in preventing osteochondrogenic transdifferentiation of VSMCs and subsequent vascular calcification in patients at high risk. 

## Figures and Tables

**Figure 1 antioxidants-09-00963-f001:**
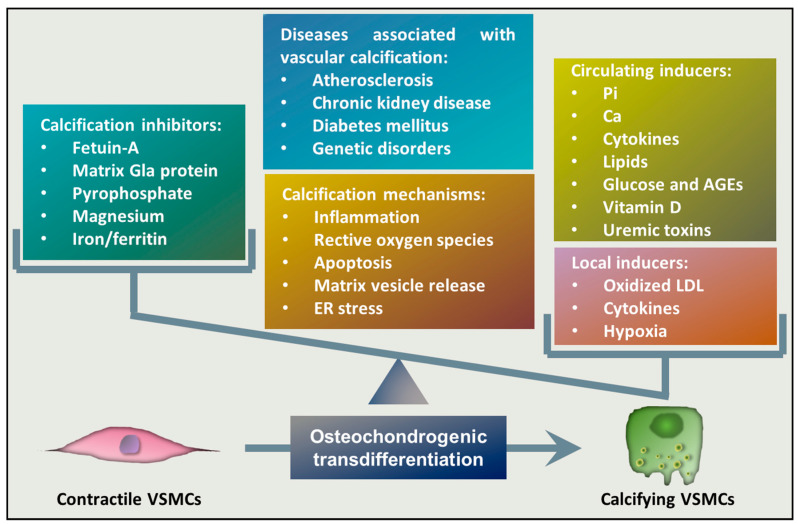
Mechanism of vascular calcification. Imbalance between calcification inhibitors and inducers is considered as a key feature in the pathology of vascular calcification. Unless repressed by calcification inhibitors, circulating and local triggers induce a transdifferentiation process of VSMCs, resulting in a phenotype switch of VSMCs from a contractile to an osteochondrogenic phenotype. The key mechanisms associated with vascular calcification include inflammation, unfettered reactive oxygen species (ROS) production, apoptosis, matrix vesicle release, and ER stress. Abbreviations: VSMCs: vascular smooth muscle cells, AGE: advanced glycation end-products, LDL: low-density lipoprotein, ER: endoplasmic reticulum.

**Figure 2 antioxidants-09-00963-f002:**
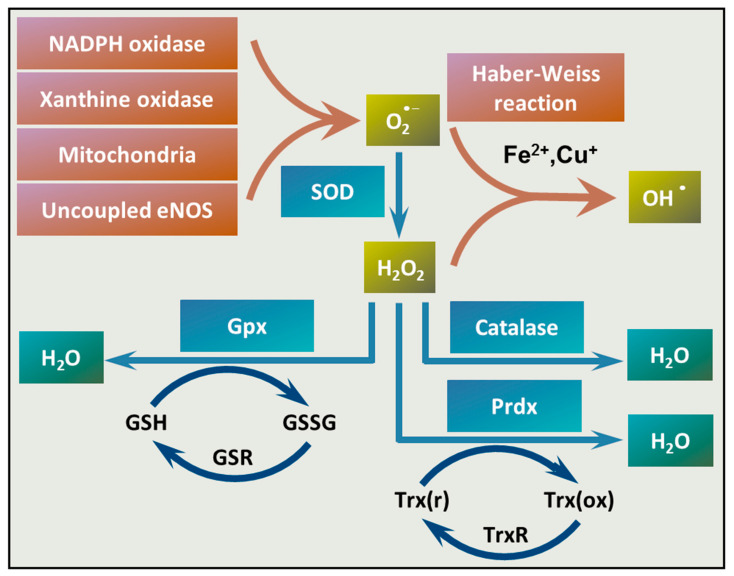
Mechanisms of ROS production and elimination in the vasculature. There are different mechanisms of ROS formation and elimination in the vasculature. NADPH oxidases, xanthine oxidases, mitochondria, or uncoupled endothelial NO synthase (eNOS) can be the source of superoxide anion (O_2_^•−^). Superoxide anion is converted to H_2_O_2_ by SOD. Transition metals, such as Fe^2+^ or Cu^+^ catalyze the reaction between O_2_^•−^ and H_2_O_2_ that yields hydroxyl radical (OH^•^) (Haber–Weiss reaction). Catalase, glutathione peroxidases (Gpx), and peroxiredoxins (Prdxs) eliminate H_2_O_2_. Abbreviations: SOD: superoxide dismutase, Gpx: glutathione peroxidase, Prdxs: peroxiredoxins, GSH: reduced glutathione, GSSG: glutathione disulfide, GSR: glutathione-disulfide reductase, NADPH: reduced nicotinamide adenine dinucleotide phosphate, Trx(r): reduced thioredoxin, Trx(ox): oxidized thioredoxin, TrxR: thioredoxin reductase.

**Figure 3 antioxidants-09-00963-f003:**
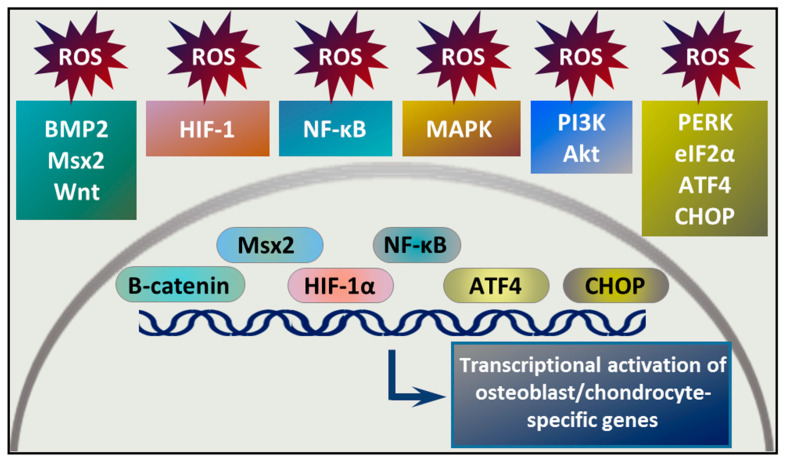
Signal transduction pathways implicated in ROS-induced osteochondrogenic phenotype switch of VSMCs. Various signal transduction pathways are involved in osteochondrogenic transdifferentiation of VSMCs. Some of these pathways are regulated by ROS or produce ROS upon activation. Activation of these pathways leads to nuclear translocation of transcription factors, such as β-catenin, Msx2, hypoxia inducible factor alpha subunit (HIF-1α), nuclear factor kappa B (NF-κB), activating transcription factor 4 (ATF4), and CCAAT-enhancer-binding protein homologous protein (CHOP). These transcription factors are involved in the transcriptional regulation of the VSMC phenotype transition from a contractile to an osteochondrogenic type.

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
