# Peer review of "Regulation of Vascular Calcification by Reactive Oxygen Species"

_antioxidants, 2020, doi:10.3390/antiox9100963_

Round 1

Reviewer 1 Report

Toth et al should be congratulated for the tremendous bibliographic work pprovided regarding this very interesting and original subject.

There are no comments regarding the datas and organization.

However, the authors should try to reduce the length of the chapters as it becomes difficult to read given the tremendous amount of information.

Added figures may help the reader to identify what datas are most relevant. 

Author Response

Dear reviewer,

Thank you very much for revising our manuscript and the positive feedback.

We agree that the text became a bit lenghty but the journal has set up a minimum word count for reviews therefore it is difficult to change.

Regarding the suggestion about increasing the number of figures: we think that the role of the figures in a review is to highlight the most important messages. We believe that the current figures are sufficient to show the key points of this review. 

Best regards,

Viktória Jeney

Reviewer 2 Report

Roles of ROS in vascular smooth muscle cell calcification are nicely summarized in this review.

The idea of treating cardiovascular diseases by antioxidants has been around for many years, but the results have been disappointing.  In this regard, it is possible that ROS is not a good target for treatment.  However, it is important to update any field of study from time to time, and in this sense, this review is of value.

Moderate English editing is needed by a native English speaking biologist/clinician (i.e. not an editor with a degree in English lit.).

Author Response

Dear Reviewer,

Thank you very much for evaluating our manuscript and the positive feedback. We agree on your point that antioxidants failed to cure cardiovascular diseases although ROS evidently play a role in them. This might be due to the huge complexity of the pathomechanism of cardiovascular diseases. 

As suggested the manuscript went through an extensive English editing. The applied changes are tracked for easier identification. 

Best regards,

Viktória Jeney

Reviewer 3 Report

A well-written review detailing the roles of oxidative stress in vascular calcification. I have no other major concerns.

The structure and flow of this review article are well-presented, and the citations are appropriate as well. Perhaps, the authors can add a small paragraph at the end of the review on future perspective on re-balancing the oxidative state in the vasculature to combat vascular calcification?   

Author Response

Dear Reviewer,

Thank you very much for evaluating our manuscript and the positive feedback. 

As suggested we compeled the Concluding remarks as the following:

"Coronary artery calcium score, a measurable indicator of vascular calcification, is an independent predictor of major adverse cardiac events in both the general population and in patients with coronary artery disease. Additionally, calcification of heart valves is the major cause of valve insufficiency.

Far from being a passive process, vascular calcification is an active and regulated process with remarkable complexity that involves numerous mechanisms that resembles bone formation. The key event in vascular calcification is the phenotype switch of vascular smooth muscle cells from contractile to osteochondrogenic phenotype. Numerous inducers of this osteochondrogenic phenotype switch triggers excess ROS production, and ROS have a causative role in the transdifferentiation process through its interactions with some major osteochondrogenic signal transduction pathways. 

So far, no therapeutic intervention exists that would specifically target vascular calcification. To develop an effective and specific anti-calcification therapy we need further understanding of the molecular mechanism of the phenotype shift of VSMCs. Re-balancing the oxidative state of the vasculature may be beneficial in preventing osteochondrogenic transdifferentiation of VSMCs and subsequent vascular calcification in patients at high risk." 

Best regards,

Viktória Jeney